# X-Ray Diffraction Study of the X-112° Y-Cut of a LiTaO_3_ Crystal Modulated by Surface Acoustic Waves

**DOI:** 10.3390/ma18225134

**Published:** 2025-11-12

**Authors:** Dmitry Roshchupkin, Dmitry Irzhak, Kirill Pundikov, Rashid Fahrtdinov, Sergey Kumanyaev, Alexey Seregin

**Affiliations:** 1Institute of Microelectronics Technology and High Purity Materials, Russian Academy of Sciences, 142432 Chernogolovka, Russia; irzhak@iptm.ru (D.I.); pundikov@iptm.ru (K.P.); rash@iptm.ru (R.F.); skumanyaev2002@mail.ru (S.K.); 2National Research Centre “Kurchatov Institute”, 123182 Moscow, Russia; seregin.a83@gmail.com

**Keywords:** LiTaO_3_ crystal, surface acoustic wave, power flow angle, power flow vector, X-ray diffraction, synchrotron radiation

## Abstract

The process of X-ray diffraction on the X-112° Y-cut of a LiTaO_3_ crystal excited by surface acoustic waves (SAW) with a wavelength of Ʌ=4
μm was studied at a synchrotron radiation source in a scheme of a double-crystal X-ray diffractometer. The sinusoidal acoustic modulation of the crystal lattice leads to the appearance of diffraction satellites on the rocking curve; the number and intensity of satellites depend on the amplitude of the SAW. Analysis of X-ray diffraction spectra allowed us to determine the velocity ($V_{SAW}=3300$ m/s) and amplitudes of the SAW. For the first time experimental investigations have demonstrated the presence of the power flow angle in the X-112° Y-cut of a LiTaO_3_ crystal, i.e., a situation where the direction of acoustic energy propagation (PFV) does not coincide with the direction of the SAW wave vector $K_{SAW}$. The measured power flow angle was PFA=0.41°. This PFA value is important for designing acoustoelectronic devices in order to reduce acoustic signal losses.

## 1. Introduction

LiTaO_3_ crystals, like LiNbO_3_ crystals, have a 3m space group symmetry and are used in acoustooptics and acoustoelectronics [1,2,3,4]. The LiTaO_3_ crystal has lower piezoelectric moduli and electromechanical coupling coefficients than LiNbO_3_, but it has the better temperature characteristics which allow the crystal to be used for high-temperature applications in acoustoelectronics. The Czochralski method was used to grow the LiTaO_3_ crystal. This crystal has a melting point of $T_{m}=1650$ °C. The temperature at which it undergoes a phase transition from ferroelectric to paraelectric is $T_{C}=665$ °C.

There are several techniques applicable for investigating the propagation of acoustic waves. Primarily, this is a method for measuring the amplitude-frequency responses by comparing the input and output HF signals in an acoustoelectronic device [5,6,7]. For this purpose, methods like scanning electron microscopy and X-ray diffraction present greater potential for investigation the process of acoustic wave propagation [8,9,10,11].

Scanning electron microscopy enables the visualization of acoustic waves in piezoelectric materials in the real-time mode, i.e., it is possible to observe the diffraction phenomena in acoustic beams (Fresnel and Fraunhofer diffractions) and the interaction of acoustic waves with crystal lattice defects, and to determine the resonance frequencies of acoustic wave excitation, wavelengths, and power flow angles [8,9,10,11]. Despite its advantages, this approach does not allow one to determine the amplitudes of acoustic waves.

The application of X-ray diffraction methods for the investigation of acoustoelectronic devices (high-resolution X-ray diffraction and X-ray topography) is of great interest, since X-ray radiation diffracts on the crystal lattice excited by the SAW. X-ray topography and X-ray diffraction are ideal methods, providing visualization of surface acoustic fields in the Fresnel diffraction region and enabling their investigation via diffraction spectrum measurements in the Fraunhofer diffraction region.

In the field of Fresnel diffraction, it is optimal to carry out research at a synchrotron radiation source, since to visualize the surface acoustic waves on the crystal surface, it is necessary to use the method of stroboscopic X-ray topography under the conditions of synchronization of the temporal structure of X-ray radiation with the excitation frequency of acoustic waves [12,13,14,15,16] or use the Talbot effect on synchrotron radiation sources with the spatial-temporal coherence of synchrotron radiation, which allows the direct observation of SAWs at the Talbot distance [17,18]. The use of these approaches allows one not only to visualize the acoustic wave field on the crystal surface and to study the interaction of SAW with the crystal lattice defects, but also to determine the SAW amplitudes, since the edge of the Fresnel diffraction region is determined by the SAW amplitude. Another important factor is the ratio of the SAW amplitude to the interplanar spacing in the crystal lattice.

Within the Fraunhofer diffraction region, the rocking curve of a crystal under acoustic modulation exhibits the diffraction satellite, owing to the fact that the sinusoidal crystal lattice modulation acts as a diffraction grating. The number and intensity of the diffraction satellites depend on the amplitude of the SAW and on the ratio of the SAW amplitude to the interplanar spacing in the crystal [19,20,21].

It should be noted that investigations using X-ray radiation require a number of conditions to be met. First of all, the substrate must have a perfect crystal structure, which allows one to obtain a rocking curve with a full width at half maximum (FWHM) close to the theoretical value. Only in this case will it be possible to observe the diffraction satellites on the rocking curve; the angular divergence between diffraction satellites must exceed the FWHM of the rocking curve. In addition, the crystal structure perfection allows for uniform reflection of X-ray radiation from the substrate surface (constant Bragg angle value). The use of synchrotron radiation is also important due to its high intensity and extremely low divergence, which makes it possible to reduce the time required to carry out the investigation. Another fundamental aspect is the possibility to obtain information about the amplitudes of the surface acoustic waves and to study the interaction of acoustic waves with the crystal lattice defects [22,23,24].

In the present work, the process of X-ray diffraction on the X-112° Y-cut of a LiTaO_3_ crystal modulated by a SAW with a wavelength of Ʌ=4 μm was studied at a synchrotron radiation source. In LiTaO_3_, crystals for acoustoelectronic devices based on SAW, two cuts are commonly used: X-112° Y-cut and 36° YX-cut. The X-112° Y-cut of a LiTaO_3_ crystal is used in SAW devices and has the SAW velocity of $V_{SAW}=3300$ m/s and square of the electromechanical coupling coefficient of $k_{SAW}^{2}=0.22$ (square of the coefficient of conversion of electrical signal into acoustic vibrations of the crystal lattice). The 36° YX-cut of a LiTaO_3_ crystal is lower in terms of the SAW velocity and square of the electromechanical coupling coefficient ($V_{SAW}=3124$ m/s and $k_{SAW}^{2}=0.05$). However, the 36° YX-cut of a LiTaO_3_ crystal is used in acoustoelectronic devices based on pseudo-surface acoustic waves (PSAW) which have high velocities ($V_{PSAW}=4224$ m/s) and a high value of the square of the electromechanical coupling coefficients ($k_{PSAW}^{2}=5.52$). Investigation of the PSAW propagation process using the X-ray diffraction method is a more complicated task, since PSAW is a wave flowing deep into the crystal. Therefore, the present work focuses only on the investigation of the SAW propagation process in the X-112° Y-cut of a LiTaO_3_ crystal.

## 2. Fabrication of SAW Delay Time Line

A congruent LiTaO_3_ crystal with a diameter of 80 mm and a length of 100 mm was grown using the Czochralski method at the Institute of Microelectronics Technology and High Purity Materials of the Russian Academy of Sciences. A surface acoustic wave delay time line consisting of two interdigital transducers (IDT) was fabricated on the surface of the X-112° Y-cut of a LiTaO_3_ crystal substrate. IDT structures were created on the crystal surface using the electron beam lithography in PMMA 950 positive resist. Next, a 1000 Å thick Al layer was deposited on the substrate surface using thermal sputtering. Following the “lift-off” process, an Al IDT structure was left on the crystal surface. The interdigital transducer consists of 25 electrode pairs and is designed to excite the SAW with a wavelength of Ʌ=4 μm. The width of the electrodes and the gap between them is 1 μm. The IDT aperture is W=60Ʌ=240 μm.

The amplitude-frequency response of the SAW delay time line was measured and is shown in Figure 1. As seen from the figure, the operational frequency band of the delay time line is 820–830 MHz. Accordingly, the central resonance frequency of the SAW excitation is $f_{0}=825$ MHz, which corresponds to the SAW velocity of $V_{SAW}=f\times Ʌ=825\times 4=3300$ m/s.

## 3. Experimental Set-Up

The study of the process of X-ray diffraction on an acoustically modulated X-112° Y-cut of a LiTaO_3_ crystal was carried out at the Kurchatov synchrotron radiation source using a double-crystal X-ray diffractometer at the optical beam line RKFM. The characteristics of the optical beam line RKFM are presented in [25]. Figure 2 shows a diagram of the experimental set-up.

After the bending magnet, the X-ray radiation was collimated by an entrance slit with a 1×1 mm^2^ aperture. A Si(111) double-crystal monochromator was used to select 10 keV X-rays (X-ray wavelength of λ=1.23 Å). A secondary 50×50 μm^2^ entrance slit then collimated the beam before it hit an acoustically modulated crystal at the Bragg incident angle. The diffracted radiation was recorded by a standard NaI scintillation detector. To measure the rocking curves, an angular scanning of the studied crystal was carried out at the angle Θ. After monochromatization and collimation, the photon flux of the X-ray beam did not exceed 10^5^ photons per second, which did not cause a change in the temperature of the LiTaO_3_ crystal and, accordingly, did not cause a change in the interplanar spacing and the value of the Bragg angle. Furthermore, no changes in the SAW resonance excitation frequency or SAW velocity were observed during the X-ray interaction with the crystal surface. The resonance frequency and velocity of SAW measured during X-ray diffraction on an acoustically modulated crystal correspond to the resonance frequency and velocity of SAW obtained based on the measurement of the amplitude-frequency response (Figure 1).

A high frequency generator Marconi (10 kHz–1 GHz) with adjustable output signal amplitude in the range of 1–12 V was used to excite the SAW.

The study of X-ray diffraction on an acoustically modulated X-112° Y-cut of a LiTaO_3_ crystal was carried out using reflection from the (110) planes. In this case, the interplanar spacing was $d_{\left(110\right)}=2.6067$ Å. Hence, a Bragg angle of $\Theta_{B}=13.645{}^{\circ}$ was obtained at an X-ray energy of E=10 keV.

SAW propagation induces the sinusoidal crystal lattice modulation (ultrasonic superlattice), resulting in the emergence of the diffraction satellites on the rocking curve. The angular divergence between the diffraction satellites is determined from the next expression [19,20,21](1)$$\delta \Theta_{mRC}=md/Ʌ,$$
with m, d, and Ʌ corresponding to the diffraction satellite index, interplanar spacing, and SAW wavelength, respectively. With an interplanar spacing of $d_{\left(110\right)}=2.6067$ Å and a SAW wavelength of Ʌ=4 μm, the calculated angle between the diffraction satellites from Expression (1) was $\delta \Theta_{mRC}=0.0037{}^{\circ}$. For diffraction satellites to be resolved on the rocking curve, their angular divergence must be greater than the FWHM of the Bragg peak. Otherwise, only a widening of the rocking curve will be observed. A direct correlation exists between the SAW amplitude and the number and intensity of the diffraction satellites; higher SAW amplitudes produce a larger number of diffraction satellites on the rocking curve [19,20,21].

An important factor affecting X-ray diffraction in an acoustically modulated crystal is the penetration depth of both the X-ray radiation and the surface acoustic wave. While the SAW penetration depth $\mu_{SAW}^{-1}$ is generally 1–2 SAW wavelengths, the penetration depth of X-rays is given by the expression [20,21](2)$$\mu_{Z}^{-1}=sin\Theta_{B}\left(E\right)/2\mu_{l}(E),$$
where $\Theta_{B}$ is the Bragg incidence angle of X-ray radiation on the crystal surface, $\mu_{l}$ is the linear absorption coefficient, and E is the X-ray energy.

Figure 3 illustrates the X-ray penetration depth in the X-112° Y-cut of a LiTaO_3_ crystal as a function of the incident radiation energy (E). At X-ray energy of E=10 keV, the X-ray penetration depth does not exceed 1 μm, since this energy corresponds to the L− edge of Ta absorption in the LiTaO_3_ crystal. Thus, at E=10 keV, the condition $\mu_{SAW}^{-1}>\mu_{Z}^{-1}$ is satisfied, meaning the acoustic wave penetrates deeper than the X-rays. However, it should be noted that the SAW amplitude and X-ray intensity decrease exponentially with crystal depth. Notably, the SAW amplitude below the surface can exceed its value on the surface [5].

## 4. X-Ray Diffraction by SAW in the X-112° Y-Cut of a LiTaO_3_ Crystal

High-resolution X-ray diffraction experiments were performed on the X-112° Y-cut of a LiTaO_3_ crystal excited by the SAW with a wavelength of Ʌ=4 μm. A high-resolution double-crystal X-ray diffractometer at the optical beam line RKFM at the Kurchatov synchrotron radiation source was used for investigation [25].

Figure 4 demonstrates the results of the high-resolution X-ray diffraction study of acoustically modulated X-112° Y-cut of a LiTaO_3_ crystal. The propagation of the surface acoustic wave induces a sinusoidal modulation of a crystal lattice, resulting in the appearance of the diffraction satellites on the rocking curve. The SAW was excited at the resonance frequency of $f_{0}=825$ MHz. This frequency aligns with the maximum of the amplitude-frequency response of the SAW delay time line, as presented in Figure 1.

Figure 4b displays the rocking curve of the crystal without SAW modulation. The FWHM of the rocking curve is 0.0025°. The calculated angular divergence between the diffraction satellites from Expression (1) is $\delta \Theta_{mRC}=0.0037{}^{\circ}$. Consequently, the angular divergence between the diffraction satellites exceeds the FWHM of the rocking curve, allowing the diffraction satellites to be clearly resolved on the rocking curve.

Figure 4a displays the dependences of the diffraction satellite intensity as a function of the amplitude of the high-frequency input signal U applied to the IDT. It should be noted that the number of the visible diffraction satellites on the rocking curve increases with the increase in the SAW amplitude, which is directly controlled by the power of the input signal applied to the IDT. The first-order satellites emerge immediately upon the application of a minimal RF-signal amplitude to the transducer that corresponds to the beginning of the SAW excitation. As the SAW amplitude is increased, the higher-order diffraction satellites become visible; for instance, the second-order satellites appear at an input signal amplitude of U = 0.6 V. As visible in the figure, the intensity of the zero diffraction peak (m=0) exhibits a non-monotonic dependence on the SAW amplitude, as it initially decreases, then increases, before falling again. This occurs because the intensity distribution among the satellites corresponds to the dependencies of the Bessel functions, where the order of the Bessel function corresponds to the order of the diffraction satellite [20,21].

The intensity of the first (m=1) diffraction satellites reaches its maximum at an input signal amplitude of U=1.5 V (Figure 4a). The corresponding rocking curve is presented in Figure 4c. Three diffraction satellites are visible on each side of the Bragg peak. Maximum intensity of the m=1 diffraction satellite attains practically 39% of the Bragg peak intensity of a nonmodulated crystal. After reaching the first maximum, the intensity of the m=1 diffraction satellite begins to decrease and reaches the second maximum when the input signal amplitude reaches U=5 V at the IDT. In this case, the intensity of the m=1 diffraction satellites also varies according to a Bessel oscillating law.

The maximal intensity of the second-order diffraction satellites (m=2) occurs at an IDT input signal amplitude of U=2.7 V (Figure 4a,d). At maximum, the intensity of the m=2 diffraction satellites reaches 23% of the intensity of the Bragg peak in the absence of the SAW. Four diffraction satellites can be observed on the rocking curve on both sides of the Bragg peak. At this input signal amplitude on the IDT  U=2.7 V, the intensity of the m=1 diffraction satellites exceeds the intensity value of the zero diffraction satellite.

The amplitude of the SAW can be defined by analyzing the diffraction spectra and using the dynamic diffraction theory [20]. This procedure requires a considerable amount of time. A simpler approach can be used with the analytical expression [26](3)$$h_{0}\approx md/2\pi ,$$
where $h_{0}$ is the SAW amplitude, m is the maximum number of the visible diffraction satellites on the rocking curve, and d is the interplanar spacing. Previously, in [26], a good agreement was demonstrated between the values of the SAW amplitude obtained from the empirical Dependence (3) and the values of the amplitude obtained from the dynamic theory of X-ray diffraction. The accuracy of the determination using the empirical dependence increases significantly with an increase in the number of diffraction satellites on the rocking curve. For example, from Expression (3) it is possible to determine the approximately SAW amplitude for Figure 4c. Three diffraction satellites are visible on the rocking curve, but in reality there are only two diffraction satellites with high intensity. Therefore, for the calculation, we can use the number of intense diffraction satellites, m=2. Then, at an interplanar spacing of $d_{\left(110\right)}=2.6067$ Å, the SAW amplitude is $h_{0}\approx 0.83$ Å. The absolute value of the SAW amplitude of $h_{0}\approx 0.83$ Å seems insignificant at first glance, but in reality it accounts for almost 30% of the interplanar spacing $d_{\left(110\right)}=2.6067$ Å. Two factors influence the number of diffraction satellites on the rocking curve. The first is related to the ratio of the SAW amplitude to the interplanar spacing, and the second is related to the ratio of the SAW amplitude to the SAW wavelength. While for the first factor the number of diffraction satellites increases with an increase in the SAW amplitude, for the second factor it is important to increase the SAW amplitude and decrease the SAW wavelength.

SAW propagation in the X-112° Y-cut of a LiTaO_3_ crystal was also studied by X-ray diffraction. Using this approach, it is possible to measure the power flow angles, defined as the angular difference between the direction of the power flow vector (PFV) and the SAW wave vector. For the investigations, the sample was placed in the angular position of the second diffraction satellite (Figure 4d, red arrow), followed by two-coordinate moving of the sample in the plane of the substrate surface.

A two-dimensional mapping of the distribution of the diffracted X-ray intensity in the second diffraction satellite on the crystal surface is presented in Figure 5. The map shows the IDT area with widening due to X-ray scattering on the wires that supply the high-frequency electrical signal to the IDT contact areas. The most interesting thing is that on the map of the distribution of diffracted X-ray radiation, one can observe the power flow angle, i.e., the angle between the PFV and the SAW wave vector $K_{SAW}$. In this case, the measured value of the power flow angle is PFA=0.41°. This PFA value is practically not visible, but it can be observed on the larger sizes of the crystal substrate. It should also be noted that there is no divergence of the acoustic beam, i.e., the width of the acoustic beam remains virtually unchanged along the crystal length.

Articles [27,28] reported the presence of the power flow angle in the X-112° Y-cut of a LiTaO_3_ crystal, but the value of the power flow angle was not given. Figure 6 schematically shows the directions of atoms displacement in the crystal lattice during the SAW propagation. The normal component of displacements $u_{1}$ determines the SAW amplitude, the longitudinal component of displacements $u_{2}$ corresponds to the direction of the SAW wave vector $K_{SAW}$, and the presence of the transverse component of displacements $u_{3}$ leads to a change in the direction of propagation of the acoustic energy. Article [29] presents the results of the calculation of displacement components $u_{1}$, $u_{2}$, and $u_{3}$ for the X-112° Y-cut of a LiTaO_3_ crystal.

The calculation of PFA is usually based on an analysis of the direction of the Umov-Poynting vector, which characterizes the energy flux density. To perform this calculation, the full suite of the crystal’s material parameters must be considered: elastic constants, piezoelectric coefficients, and dielectric permeability of the crystal. Currently, several sets of material constants are known and used in calculations for LiTaO_3_ crystals [30,31,32,33].

Table 1 compiles the results from measuring and calculating of the SAW velocity and power flow angle for the X-112° Y-cut of a LiTaO_3_ crystal using different sets of material constants.

The results of the calculation of the SAW velocity and PFA using the set of material constants from [30] coincide with the results of measurements presented in [34], but there is a small mismatch with the measurements of the SAW velocity and PFA in this work using the X-ray diffraction method. In articles [31,32,33], there is a more significant difference in the SAW velocities and power flow angles from the results obtained in these studies.

A slight difference in the SAW velocity is not a critical parameter affecting the accuracy of SAW devices, as it can be corrected by selecting the operating frequency or the thickness of the metal electrodes. At the same time, accuracy in the measurement of the power flow angle is extremely important. Firstly, non-zero PFV causes the wavefront to become distorted, which reduces the efficiency of interaction between the elements of the acoustic tract and the acoustic wave, increasing signal loss. Secondly, for high-frequency applications (few GHz), where the length of the acoustic wave becomes compatible with the dimensions of the structural elements of the acoustic tract, the influence of PFA becomes critically important. Thus, it is necessary to verify the set of material constants, since when developing the high-frequency SAW devices, any deviation of the PFA in the crystal from the calculated values can lead to incorrect operation of the device.

If the power flow angle is PFA=0.41° in the X-112° Y-cut of a LiTaO_3_ crystal, then at a distance of 10 mm, the acoustic beam displacement will be 73 μm, which is not a very large distance and will not be noticeable at large SAW wavelengths. The deviation of the acoustic beam leads to increased losses in the acoustoelectronic devices, since not all of the acoustic beam enters the output IDT aperture. In principle, the existence of the power flow angle is not important in the fabrication of the SAW-resonators, but it is an important aspect in the creation of SAW filters and delay time lines. When creating the latter, it is necessary to take into account the power flow angle and displace the output IDT from the axis of the SAW wave vector by the required distance, which reduces the acoustic signal losses. However, when the wavelength of the SAW is reduced to 10 μm or less, this shift will already be noticeable. Let us consider the visualization of the propagation process of a traveling SAW in the X-112° Y-cut of a LiTaO_3_ crystal using scanning electron microscopy, which allows one to observe the acoustic wave fields on the crystal surface in the real-time mode and to determine the power flow angles [8,9,10,11].

Figure 7 shows a SEM micrograph of a traveling SAW propagation on the surface of the X-112° Y-cut of a LiTaO_3_ crystal. The traveling SAW with a wavelength of Ʌ = 60 μm propagates with a velocity of $V_{SAW}=3300$ m/s. At a small power flow angle and a large SAW wavelength, the change in the direction of the power flow vector is practically invisible. The wavefront is rounded at the edges of the acoustic beam. However, reducing the wavelength of the SAW will allow one to visualize the power flow angle.

However, there are materials with high values of the power flow angle. Consider, for example, the X-cut of a La_3_Ga_5_SiO_14_ crystal, which has a large value of a power flow angle. Figure 8 shows a SEM micrograph of a traveling SAW with a wavelength of Ʌ=10 μm, which propagates with a velocity of $V_{SAW}=2470$ m/s in the X-cut of a La_3_Ga_5_SiO_14_ crystal. The resonance frequency of the SAW excitation is $f_{0}=247$ MHz. The photograph shows the angular deviation of the power flow vector with a power flow angle of PFA=6.4°. It should also be noted that the SAW excited at the input IDT travels past the output IDT, i.e., in this case, the acoustic signal is completely lost.

Thus, the use of different methods of diagnostics is an important aspect for the investigation of the acoustoelectronic devices based on the surface acoustic waves.

## 5. Conclusions

For the first time the process of SAW propagation in the X-112° Y-cut of a LiTaO_3_ crystal was investigated at the synchrotron radiation source using high-resolution X-ray diffraction. The measurements of the rocking curves of acoustically modulated crystals were used to determine the SAW amplitude, resonance excitation frequency, and power flow angle. X-ray radiation is sensitive to crystal lattice distortions due to SAW propagation.

To determine the optimal excitation parameters, rocking curves were recorded at various SAW frequencies. Analysis of rocking curves showed that SAW resonance excitation frequency is achieved at a frequency of $f_{0}=825$ MHz, as evidenced by the maximum number of diffraction satellites and intensity. This frequency value corresponds to the surface acoustic wave propagation velocity $V_{SAW}=3300$ m/s. The measured resonance frequency of the SAW excitation corresponds to the center frequency of the amplitude-frequency response of the SAW delay time line under study (Figure 1).

Furthermore the investigation of the dependence of the diffraction satellite intensities versus amplitude of the input signal on the IDT was carried out. The obtained results show that a higher amplitude of the input signal at the IDT leads to the appearance of a greater number of the diffraction satellites on the rocking curve. The dependencies of the diffraction satellite intensity on the input signal amplitude at the IDT were obtained. Within the input voltage range of U=2−2.7 V applied to the IDT, the intensity of the first diffraction satellite exceeds the intensity of the zero diffraction satellite. At an input signal amplitude of U=1.5 V, the intensity of the first diffraction satellite reaches its maximum value, which is practically 39% of the intensity of the zero Bragg peak in the absence of acoustic modulation.

The power flow angle of PFA=0.41° was measured in the X-112° Y-cut of a LiTaO_3_ crystal.

Thus, the possibility of using the double-crystal X-ray diffraction method at a synchrotron radiation source for analyzing the SAW propagation process in the X-112° Y-cut of a LiTaO_3_ crystal has been demonstrated.

## Figures and Tables

**Figure 1 materials-18-05134-f001:**
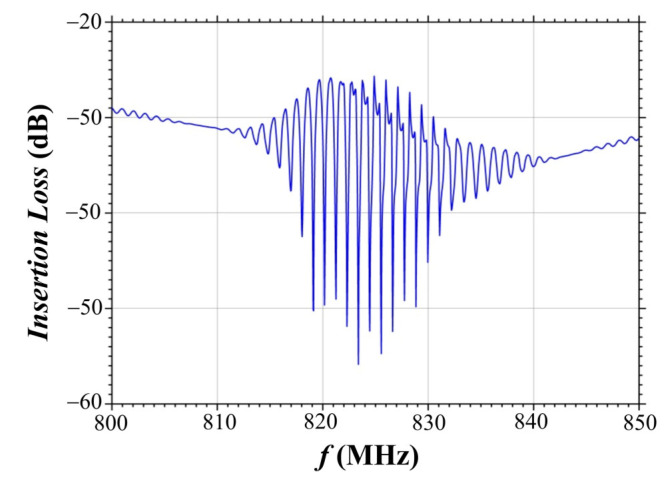
Amplitude-frequency response of the SAW delay time line.

**Figure 2 materials-18-05134-f002:**
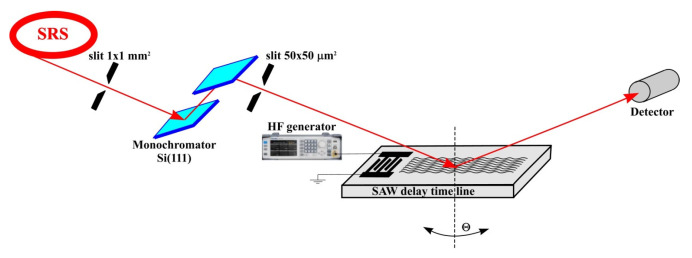
Double-crystal X-ray diffractometer set-up.

**Figure 3 materials-18-05134-f003:**
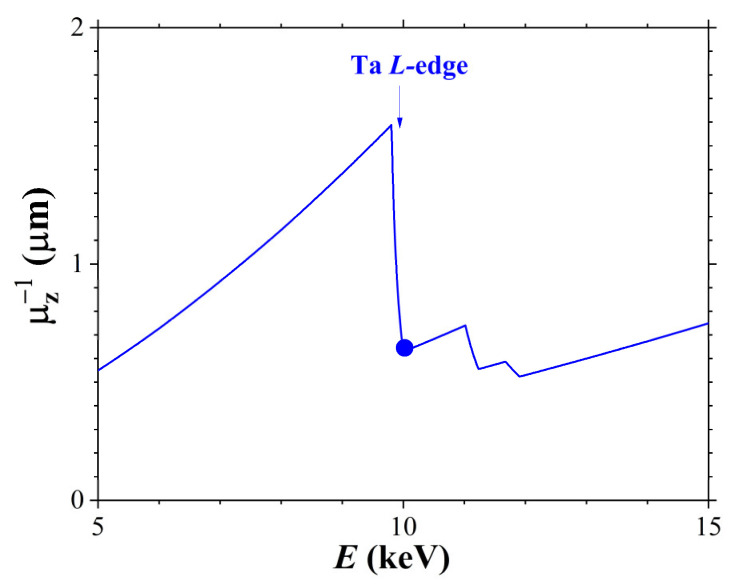
X-ray penetration depth versus X-ray energy for the (110) reflection in a LiTaO_3_ crystal. The circle denotes the value at 10 keV.

**Figure 4 materials-18-05134-f004:**
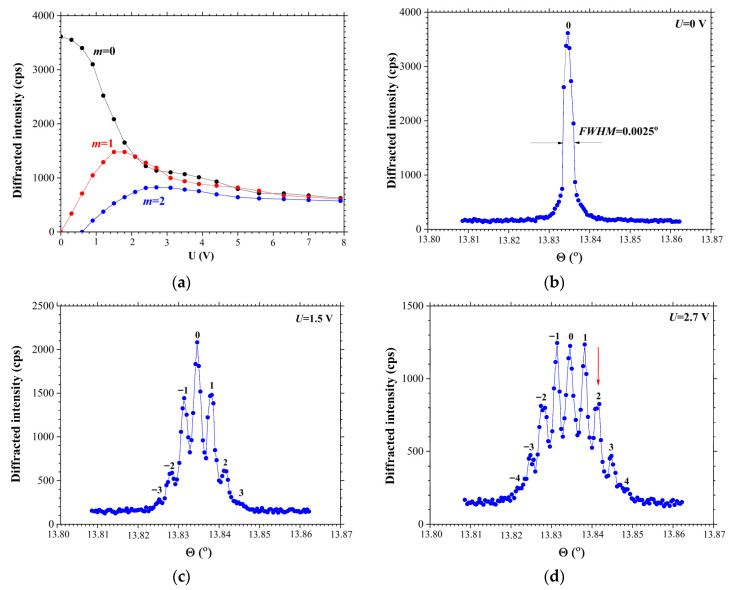
X-ray diffraction on a SAW-modulated LiTaO_3_ crystal (X-112° Y-cut, Λ=4 µm). (**a**) Diffraction satellite intensity as a function of the input signal amplitude on the IDT *U*. Rocking curves 
of the (110) reflection at E=10 keV: (**b**) without SAW excitation (U=0 V), (**c**) SAW excitation at U=1.5 V, (**d**) U=2.7 V.

**Figure 5 materials-18-05134-f005:**
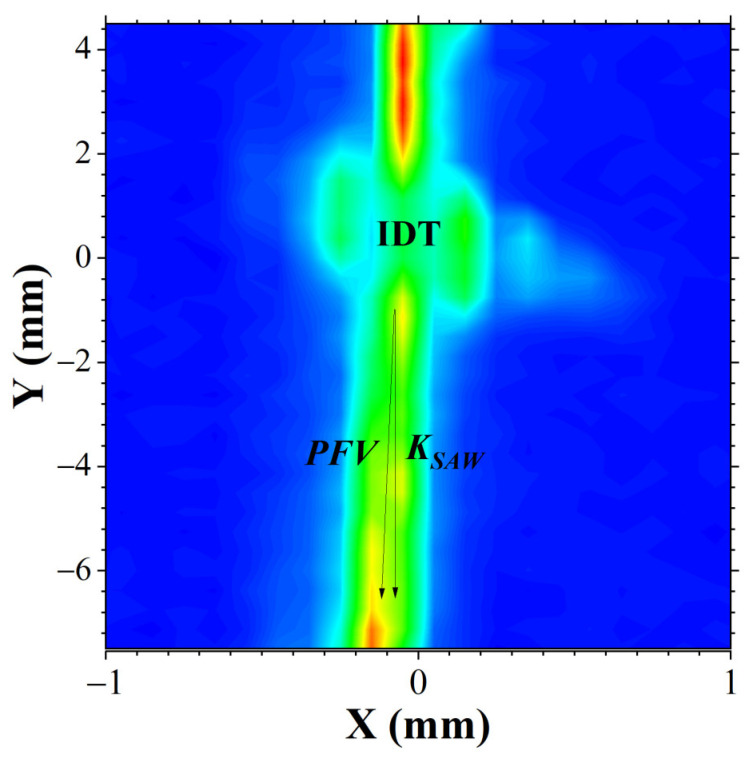
Map of the diffracted X-ray intensity on the surface of the X-112° Y-cut of a LiTaO_3_ crystal under SAW modulation. Reflection (110), X-ray energy E=10  keV.

**Figure 6 materials-18-05134-f006:**
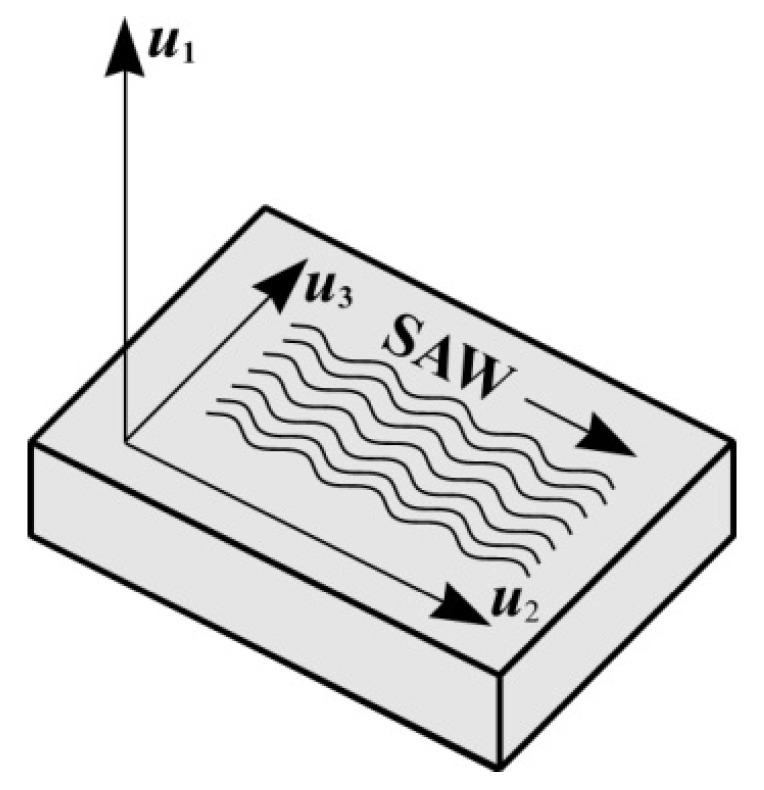
Scheme of the normal ($u_{1}$
), longitudinal ($u_{2}$), and transverse components ($u_{3}$) of the crystal lattice deformation at the process of the SAW propagation.

**Figure 7 materials-18-05134-f007:**
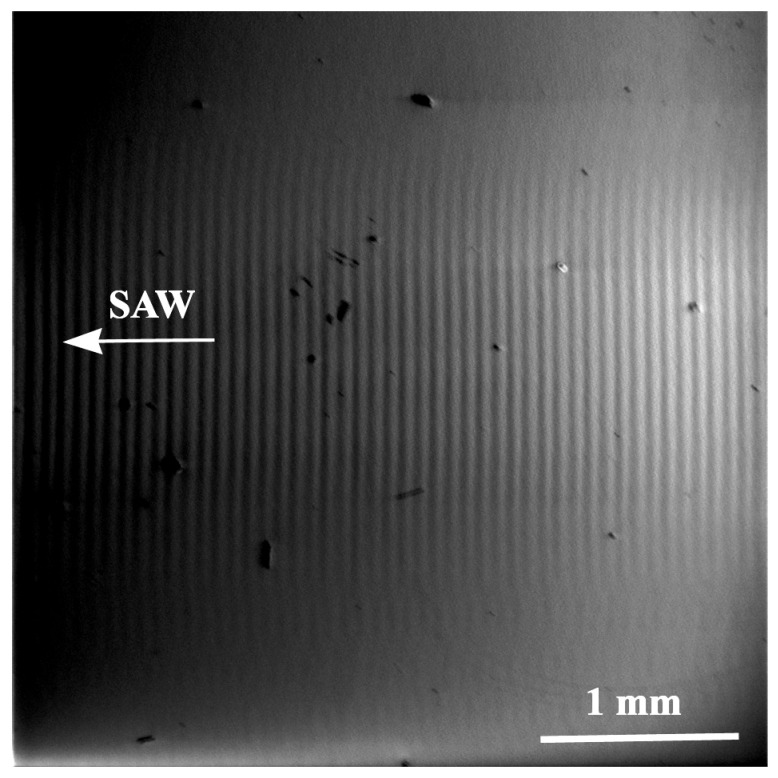
SEM image of the SAW propagation in the X-112° Y-cut of a LiTaO_3_ crystal modulated by the SAW with a wavelength of Ʌ=60
µm. $V_{SAW}=3300$ m/s, $f_{0}=55$ MHz.

**Figure 8 materials-18-05134-f008:**
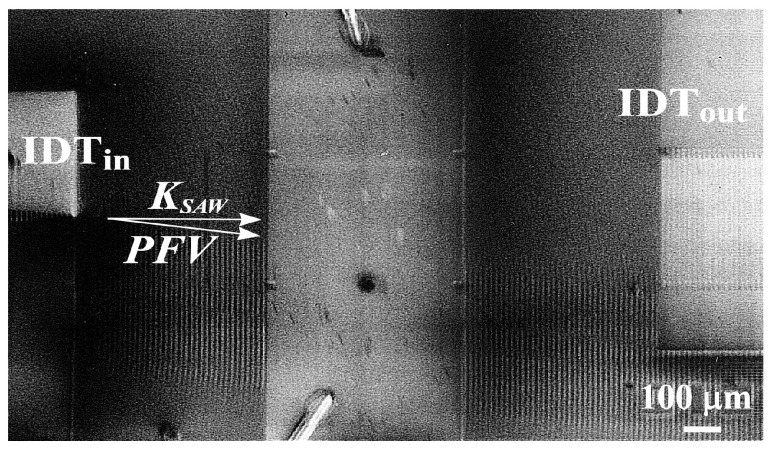
SEM image of the SAW propagation in the X-cut of a La_3_Ga_5_SiO_14_ crystal modulated by the SAW with a wavelength of Ʌ=10
µm. $V_{SAW}=2470$ m/s, $f_{0}=247$ MHz.

**Table 1 materials-18-05134-t001:** Results of measurements and calculations of PFA in the X-112° Y-cut of a LiTaO_3_ crystal.

	Measurements, Present Article	Calculations, [30]	Calculations, [31]	Calculations, [32]	Calculations, [33]	Measurements, [34]
$V_{SAW}$ (m/s)	3300	3301	3295	3307	3320	3301
PFA (°)	0.41	0.48	0.04	1.5	0.25	~0.5

## Data Availability

The original contributions presented in this study are included in the article. Further inquiries can be directed to the corresponding author.

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
