# Peer review of "X-Ray Diffraction Study of the X-112° Y-Cut of a LiTaO3 Crystal Modulated by Surface Acoustic Waves"

_materials, 2025, doi:10.3390/ma18225134_

Round 1

Reviewer 1 Report

Comments and Suggestions for Authors

Roshchupkin et al present a study of X-ray diffraction on X-112° Y -Cut of a LiTaO3 Crystal, and also employ the electron microscopy technique. The study is detailed, and I would recommend minor publication after these suggestions.

  1. It's better to mention the space group than mention the point group symmetry. Authors are recommended to change it in the first line of the introduction.
  2. It's unclear if the LiTaO3 crystals were synthesized in the laboratory or were purchased. Authors need to mention it in the experimental section.
  3. Enlarge the axis labels in all figures.

Author Response

Dear Reviewer,

Thank you for your useful comments.

Comments 1: It's better to mention the space group than mention the point group symmetry. Authors are recommended to change it in the first line of the introduction.

Response 1: We replaced “point group symmetry” by “space group symmetry”/

Comments 2: It's unclear if the LiTaO3 crystals were synthesized in the laboratory or were purchased. Authors need to mention it in the experimental section.

Response 2: A congruent LiTaO3 crystal with a diameter of 80 mm and a length of 100 mm was grown using the Czochralski method at the Institute of Microelectronics Technology and High Purity Materials of the Russian Academy of Sciences.

Comments 3: Enlarge the axis labels in all figure

Response 3: Axis labels are enlarged in all figures.

Reviewer 2 Report

Comments and Suggestions for Authors
  1. The novelty of this work is somewhat unclear since similar XRD–SAW studies by the same group (e.g., Roshchupkin et al., J. Appl. Cryst., 2021; Materials Lett., 2024) are already cited. The authors should explicitly highlight what new findings or improvements this paper provides.
  2. The term “acoustic energy flow dissipation” is confusing. The paper later clarifies that the power flow vector does not align with the wave vector, but the word “dissipation” implies energy loss, which may mislead readers. Please rephrase this concept to “angular deviation of the power flow vector” or “non-collinearity between PFV and KSAW.
  3. The authors should include in the experimental section: The exact beamline name and parameters at the Kurchatov synchrotron (flux, beam size, monochromator details); Temperature control conditions during XRD measurement (since LiTaO₃ has strong temperature-dependent piezoelectric properties); The method of voltage calibration for the IDT and how acoustic amplitude was measured or estimated (beyond the empirical relation).
  4. Equation (3) is too simplistic and may introduce significant uncertainty. The authors should clarify whether this estimation aligns with the dynamic diffraction theory from Ref. [20] and indicate its error margin.
  5. The reported SAW amplitude of 0.83 Å seems very small. Please discuss its physical significance and how it compares with values reported for LiNbO₃ or similar crystals.
  6. The measured PFA=0.41° agrees with prior theoretical predictions, but the discussion is shallow. Please discuss how this value impacts the device performance, beam focusing, and loss mechanisms, especially in GHz-frequency applications.
  7. Figures 1–8 are well-chosen, but several need improvement in quality and clarity: Add axis labels, units, and scales to all figures (especially Figs. 4–5). Figures 7–8 (SEM images) lack scale bars and contrast enhancement; also, ensure proper permissions if reused from previous works.
  8. The manuscript requires language polishing. Many sentences are overly long or grammatically inconsistent (e.g., “which allows the crystal to be used for high-temperature applications in acoustoelectronics and sensor technology” can be shortened).
  9. Consistency in notation: “𝑋 − 112° 𝑌 −cut” vs “X–112° Y–cut” should be standardized.
  10. The abstract is clear but could include numerical results (e.g., SAW amplitude, PFA, and resonance frequency) for better impact.
  11. Ensure all references follow the Materials journal format. Re-check the DOI and year formatting in Refs. [1–30]; several are missing DOIs or have inconsistent punctuation. Consider citing recent related studies (e.g., Appl. Phys. Lett. 2023, Ultrasonics 2022) to show broader relevance.
  12. Replace “Å” with “Ångström (Å)” in text consistently.
  13. Some formula symbols are not properly formatted (e.g., 𝑉𝑆𝐴𝑊, 𝑘𝑆𝐴𝑊²). Use consistent italic/scientific notation.

Author Response

Dear Reviewer,

Thank you for your useful comments.

Comments 1: The novelty of this work is somewhat unclear since similar XRD–SAW studies by the same group (e.g., Roshchupkin et al., J. Appl. Cryst., 2021; Materials Lett., 2024) are already cited. The authors should explicitly highlight what new findings or improvements this paper provides.

Response 1: Analysis of X-ray diffraction spectra allowed us to determine the velocity ( m/s) and amplitudes of the SAW. For the first time experimental investigations also have demonstrated the presence of the power flow angle in the X-112° Y-cut of a LiTaO3 crystal, i.e., a situation where the direction of acoustic energy propagation () does not coincide with the direction of the SAW wave vector . The measured power flow angle was . This  value is important to design acoustoelectronic devices in order to reduce acoustic signal losses.

To determine the optimal excitation parameters, rocking curves were recorded at various SAW frequencies. Analysis of rocking curves showed that SAW resonance excitation frequency is achieved at a frequency of  MHz, as evidenced by the maximum number of diffraction satellites and intensity. This frequency value corresponds to the surface acoustic wave propagation velocity  m/s.

The use of X-ray diffraction to study the SAW propagation process provides reliable information (SAW velocity and amplitude, PFA), since X-ray radiation is sensitive to the changes in the parameters of the crystal unit cell. Previously, we used X-ray diffraction to measure the piezoelectric moduli in crystals of the langasite family (D. Irzhak and D. Roshchupkin, Measurement of independent piezoelectric moduli of Ca3NbGa3Si2O14, La3Ga5.5Ta0.5O14 and La3Ga5SiO14 single crystals, J. Appl. Cryst. (2018). 51, 1174–1181, https://doi.org/10.1107/S1600576718009184). Now, these piezomodule values are used to design the acoustoelectronic devices.

Comments 2: The term “acoustic energy flow dissipation” is confusing. The paper later clarifies that the power flow vector does not align with the wave vector, but the word “dissipation” implies energy loss, which may mislead readers. Please rephrase this concept to “angular deviation of the power flow vector” or “non-collinearity between PFV and KSAW.

Response 2: The necessary correction has been made in the text.

For example: Using this approach, it is possible to measure the power flow angles, defined as the angular difference between the direction of the power flow vector () and the SAW wave vector.

Comments 3: The authors should include in the experimental section: The exact beamline name and parameters at the Kurchatov synchrotron (flux, beam size, monochromator details); Temperature control conditions during XRD measurement (since LiTaO₃ has strong temperature-dependent piezoelectric properties); The method of voltage calibration for the IDT and how acoustic amplitude was measured or estimated (beyond the empirical relation).

Response 3: The study of the process of X-ray diffraction on an acoustically modulated X-112° Y-cut of a LiTaO3 crystal was carried out at the Kurchatov synchrotron radiation source using a double-crystal X-ray diffractometer at the optical beam line RKFM. The characteristics of the optical beam line RKFM are presented in [25].

[25] Kohn, V.G.; Prosekova, P.A.; Seregina, A. Yu.; Kulikova, A.G.; Pisarevsky, Yu.V.; Blagova, A.E.; Kovalchuk M.V. Experimental Study of Two-Beam X-Ray Diffractometry Using Synchrotron Radiation. Crystallography Reports 2019, 64, 24.

After monochromatisation and collimation, the photon flux on the crystal surface did not exceed 105 photons per second, which did not cause a change in the temperature of the LiTaO3 crystal and, accordingly, did not cause a change in the interplanar spacing and the value of the Bragg angle. Furthermore, no changes in the SAW excitation resonance frequency or SAW velocity were observed during the X-ray interaction with the crystal surface. The resonance frequency and velocity of SAW measured during X-ray diffraction on an acoustically modulated crystal correspond to the resonance frequency and velocity of SAW obtained based on the measurement of the amplitude-frequency response (Fig. 1).

A high frequency generator Marconi (10 kHz ÷ 1 GHz) with adjustable output signal amplitude in the range of 1÷12 V was used to excite the SAW.

Previously, in [26], a good agreement was demonstrated between the values of the SAW amplitude obtained from the empirical dependence (3) and the values of the amplitude obtained from the dynamic theory of X-ray diffraction [20]. The accuracy of the determination using the empirical dependence increases significantly with an increase in the number of diffraction satellites on the rocking curve.

The X-ray method for determining the SAW amplitude is the most accurate compared to other methods. The diffraction of laser beam has a significantly larger error, since the number of diffraction satellites is significantly smaller compared to the diffraction of X-ray radiation. The X-ray topography method can also be used to determine SAW amplitude, but this approach can only be implemented for SAW wavelengths larger than 10 μm, which is related to the parameters of 2D detectors. At the end of the year, an experiment is planned to visualize the acoustic wave fields at the synchrotron radiation source, where the plane of the direst SAW imaging is determined by the amplitude and SAW wavelength.

Comments 4: Equation (3) is too simplistic and may introduce significant uncertainty. The authors should clarify whether this estimation aligns with the dynamic diffraction theory from Ref. [20] and indicate its error margin.

Response 4: Previously, in [26], a good agreement was demonstrated between the values of the SAW amplitude obtained from the empirical dependence (3) and the values of the amplitude obtained from the dynamic theory of X-ray diffraction [20]. The accuracy of the determination using the empirical dependence increases significantly with an increase in the number of diffraction satellites on the rocking curve.

Comments 5: The reported SAW amplitude of 0.83 Å seems very small. Please discuss its physical significance and how it compares with values reported for LiNbO₃ or similar crystals.

Response 5: The absolute value of the SAW amplitude of  Å seems insignificant at first glance, but in reality, it accounts for almost 30% of the interplanar spacing  Å. Two factors influence on the number of diffraction satellites on the rocking curve. The first is related to the ratio of the SAW amplitude to the interplanar spacing, and the second is related to the ratio of the SAW amplitude to the SAW wavelength. While for the first factor the number of diffraction satellites increases with an increase in the SAW amplitude, for the second factor it is important to increase the SAW amplitude and decrease the SAW wavelength.

Comments 6: The measured PFA=0.41° agrees with prior theoretical predictions, but the discussion is shallow. Please discuss how this value impacts the device performance, beam focusing, and loss mechanisms, especially in GHz-frequency applications.

Response 6: The deviation of the acoustic beam leads to increased losses in the acoustoelectronic devices, since not all of the acoustic beam enters the IDT aperture. In principle, the existence of the power flow angle is not important in the fabrication of the SAW resonators, but it is an important aspect in the creation of SAW filters and delay time lines. When creating the latter, it is necessary to take into account the power flow angle and displace the output IDT from the axis of the SAW wave vector by the required distance, which reduces the acoustic signal losses.

Comments 7: Figures 1–8 are well-chosen, but several need improvement in quality and clarity: Add axis labels, units, and scales to all figures (especially Figs. 4–5). Figures 7–8 (SEM images) lack scale bars and contrast enhancement; also, ensure proper permissions if reused from previous works.

Response 7: We have corrected figures. SEM photographs are published for the first time. We have significantly increased the contrast in Fig. 7. In Fig. 8, we also increased the contrast, but it cannot be increased significantly, since the piezomodules in La3Ga5SiO14 crystals are significantly smaller than in LiTaO3 crystals.  SEM images are images of potential contrast, which corresponds to the distribution of the potential between the minima and maxima of the SAW. Accordingly, the potential of the minima and maxima in La3Ga5SiO14 is significantly lower than in LiTaO3.

Comments 8: The manuscript requires language polishing. Many sentences are overly long or grammatically inconsistent (e.g., “which allows the crystal to be used for high-temperature applications in acoustoelectronics and sensor technology” can be shortened).

Response 8: The text of the article has been corrected.

Comments 9: Consistency in notation: “? − 112° ? −cut” vs “X–112° Y–cut” should be standardized.

Response 9: We have standardized the notation as "X-112° Y-cut".

Comments 10: The abstract is clear but could include numerical results (e.g., SAW amplitude, PFA, and resonance frequency) for better impact.

Response 10: Analysis of X-ray diffraction spectra allowed us to determine the velocity ( m/s) and amplitudes of SAW. Studies also have demonstrated the presence of power flow angle in the X-112° Y-cut of a LiTaO3 crystal, i.e., a situation where the direction of acoustic energy propagation () does not coincide with the direction of the SAW wave vector . The measured power flow angle was .

Comments 11: Ensure all references follow the Materials journal format. Re-check the DOI and year formatting in Refs. [1–30]; several are missing DOIs or have inconsistent punctuation. Consider citing recent related studies (e.g., Appl. Phys. Lett. 2023, Ultrasonics 2022) to show broader relevance.

Response 11: We formatted all references in accordance with the format of the journal Materials.

Another fundamental aspect is the possibility to obtain information about the amplitudes of the surface acoustic waves and to study the interaction of acoustic waves with defects in the crystal lattice [22-24].

[22] Topaltzikis, D.; Wielunski, M.; Hörner, A.L.; Küß, M.; Reiner, A.; Grünwald, T.; Schreck, M.; Wixforth, A.; Rühm, W. Detection of x rays by a surface acoustic delay line in contact with a diamond crystal. Appl. Phys. Lett. 2021, 118, 133501

[23] Holstad, T.S.; Dresselhaus- Marais, L.E.; Ræder, T.M.; Kozioziemski, B.; van Driel, T.; Seaberg, M.; Folsom, E.; Eggert, J.H.; Knudsen, E.B.; Nielsen, M.M.; Simons, H.; Haldrup, K.; Poulsen, H.F. Real-time imaging of acoustic waves in bulk materials with X-ray microscopy. PNAS 2023, 120, e2307049120.

[24] Zhou, T.; Reinhardt, A.; Bousquet, M.; Eymery, J.; Leake, S.; Holt, M.V.; Evans, P.G.; Schülli, T. High-resolution high-throughput spatiotemporal strain imaging reveals loss mechanisms in a surface acoustic wave device. Nature Communications 2025, 16, 2822.

Comments 12: Replace “Å” with “Ångström (Å)” in text consistently.

Response 12: Replaced with “Å”

Comments 13: Some formula symbols are not properly formatted (e.g., ????, ????²). Use consistent italic/scientific notation.

Response 13: We formatted the formula symbols in the text and highlighted them in red.

Round 2

Reviewer 2 Report

Comments and Suggestions for Authors

I do not have any comments now, the present form of manuscript can be potential for publication.